# Deep Learning-Based Wrist Vascular Biometric Recognition

**DOI:** 10.3390/s23063132

**Published:** 2023-03-15

**Authors:** Felix Marattukalam, Waleed Abdulla, David Cole, Pranav Gulati

**Affiliations:** Department of Electrical, Computer and Software Engineering, The University of Auckland, Auckland 1010, New Zealand

**Keywords:** biometrics, wrist vein, deep learning, machine learning, Siamese Neural Network, convolutional neural network

## Abstract

The need for contactless vascular biometric systems has significantly increased. In recent years, deep learning has proven to be efficient for vein segmentation and matching. Palm and finger vein biometrics are well researched; however, research on wrist vein biometrics is limited. Wrist vein biometrics is promising due to it not having finger or palm patterns on the skin surface making the image acquisition process easier. This paper presents a deep learning-based novel low-cost end-to-end contactless wrist vein biometric recognition system. FYO wrist vein dataset was used to train a novel U-Net CNN structure to extract and segment wrist vein patterns effectively. The extracted images were evaluated to have a Dice Coefficient of 0.723. A CNN and Siamese Neural Network were implemented to match wrist vein images obtaining the highest F1-score of 84.7%. The average matching time is less than 3 s on a Raspberry Pi. All the subsystems were integrated with the help of a designed GUI to form a functional end-to-end deep learning-based wrist biometric recognition system.

## 1. Introduction

In a data-driven world, data security, privacy security and protection of personal identification information are essential. Personal Identification Numbers (PINs) and passwords are extensively used for human identification and verification. With the advancement of technology, it has become more challenging to safeguard confidential information. PINs and passwords are susceptible to spoofing, and are likely to be stolen and transferred between people [1]. Another reason is these methods can be forged as they rely on one’s memory. Other systems that use access cards and identity cards are vulnerable and can easily fail as their identity components can be misplaced or misused.

Biometric recognition systems utilise physiological or behavioural features for recognition [2]. Behavioural features include voice, signature, gait and keystroke dynamics. Physiological features include face, fingerprint, palmprint, iris and vascular patterns. There is a recent trend in more use of physiological and behavioural features for identification and verification purposes in a recognition system. The physiological methods are preferred because of their uniqueness, permanence and accuracy. Common biometric recognition systems available today utilise extrinsic features such as the face, iris and fingerprint [3]. Fingerprint is easy to spoof due to its extrinsic nature and also requires physical contact with the sensor [4]. This is a disadvantage as it is a contact-based method that can contribute to the transmission of different germs and viruses through the use of this technology. This is of more relevance after the advent of the COVID-19 pandemic. Another extrinsic feature used is face. Face recognition is quite successful as it is accurate, contactless and easy to use. However, with the advancement in the technology of face acquisition devices, it is possible to capture this feature without the consent and knowledge of the user by placing high-resolution cameras at faraway distances. Iris recognition is another example that is precise but needs a constrained environment and the acquisition device, i.e., iris scanner, is expensive.

The other type of physiological feature that can be used for biometric recognition is an intrinsic feature. Features used for biometrics are also referred to as biometric modalities. Intrinsic modalities, specifically vein modalities, are much harder to spoof [1], are mostly invisible to the human eye, do not change over time and are unique to each person [5]. Vein biometrics are classified based on the region of vein structure that is used for recognition, with the most common being palm and finger vein recognition. Technologies developed using vein as the modality are referred to as vascular biometric technologies. In many cases, they are contactless making them preferred in clinical applications. Vein information lies beneath the skin surface and is robust as it depends on the blood flow in the vein, which prevents latency and makes it difficult to forge. Moreover, veins are time-invariant in structure. The changes in the vascular system start with the development of a vascular tree in the embryonic phase. After the formation of the veins, only their spatial size increases as human beings grow [4]. Vascular biometric systems use the vein pattern, which is stable over time and hence considered robust.

The wrist vein is known to have sufficient unique features to be used in a vascular biometric system [6]. Wrist vein recognition has seen less research when compared to its palm and finger vein counterparts. However, it is of interest due to the thickness of the skin around the wrist being thin. This makes the wrist vein more accessible in the image acquisition stage of the biometric system [7].

In biometric systems, identification and authentication are often interchangeably used. It is to be noted that these terms have different meanings and significance and are dependent on the operating mode of the biometric system. The authentication step is a one-to-one matching process for a specific user where the system compares the input preprocessed image obtained from the user, also called probe image, to the database image previously stored in the system, also called the reference image. Here, the user has already claimed a specific identity and hence the reason the operating mode has only one-to-one comparison operation. This comparison is to verify the claimed identity and grant access depending on the matching output. Identification is the step where the input image is taken from the user (probe image) and is compared to all the images which are already stored in the database using the registration process. This is a one-to-N matching operational mode.

Systems for both the above-mentioned operating modes can be designed using traditional signal processing methods and deep learning methods. Deep learning has seen success in palm and finger vein recognition recently. Deep learning has the advantage of being able to encompass all the signal processing steps required for vein recognition to provide an end-to-end recognition system. Although most palm and finger vein research now focuses on deep learning [8], it has seen limited application in wrist vein recognition. A low-cost wrist vein image acquisition device was developed to capture high-quality vein pattern images. Concurrently, algorithms for image segmentation and matching were developed using the FYO database [9]. This database helped develop the segmentation and matching algorithms, after which they were integrated with the developed hardware to form an end-to-end wrist vein vascular biometric system presented in this paper.

The contributions of this paper are listed below:Extension of the literature survey carried out in our previous study [6].Segmentation of wrist vein images using a modified UNet architecture.Development of a matching engine that can compare probe image with reference image using Convolutional Neural Network (CNN) followed by Siamese Neural Network [10] for vascular biometrics.Development of a Graphical User Interface (GUI) and integration of the subsystems to form a complete end-to-end deep learning-based wrist vein biometric system.

To the best of our knowledge, this is the first study where an entire end-to-end wrist vein recognition system using a deep learning algorithm has been developed. Section 2 provides a comprehensive literature review of state-of-the-art wrist vein recognition systems and deep learning approaches applied to other vein biometrics. Section 3 presents the proposed subsystems. Section 4 reports and discusses the obtained results with its performance evaluation. Section 5 is dedicated to the conclusion and future work.

## 2. Literature Review

This section provides a review of the state-of-the-art wrist vein recognition systems covering the literature on image acquisition, databases, segmentation algorithms and matching engines. Deep learning approaches applied to vein recognition have also been reviewed for the sake of completeness.

Acquiring an image of the vein patterns is the first step in the vein recognition process. Most commercial cameras have IR filters embedded to prevent IR light from entering the sensor to avoid having unnatural-looking images. For biometric recognition, this filter needs to be removed as IR light is necessary to capture the veins. To capture an optimum image, the distance between the wrist and the camera lens, as well as the camera quality/image size needs to be accounted for. Crisan et al. suggest utilising a tight optical spectrum window of 740 nm to 760 nm to provide maximum penetration in vein biometrics [11]. In [12], Raghavendra et al. developed a low-cost vein recognition sensor system, utilised it to develop a wrist vein database comprising of 1000 images and evaluated the system with nine different feature extraction algorithms. In their design, an NIR monochrome CMOS camera was utilised with 40,870 nm NIR LEDs illuminating the wrist. When tested with the LG-SRC feature extraction technique, an equal error rate (EER) [13] of 1.63% was achieved. Pegs were used, hence making the system contact-based. Moreover, the database collected does not appear to be publicly available. A similar system was proposed in [14] by Pascual et al. A commercial dome light with 112,880 nm NIR LEDs was used to homogenise the light and eliminate shadows. There is no evidence of testing the captured images using any algorithms. In [15], Garcia-Martin et al. propose a small and simple system consisting of a Logitech webcam with filter manually removed, a custom ring PCB for NIR LEDs of 850 nm and a Raspberry Pi Model 3B. However, the webcam does not capture the contrast between skin and vein well and there is no mention of using any feature extraction algorithms either.

Although NIR images provide much clearer vein structures, using regular light has also been considered in some studies. In [16], Kurban et al. proposed using a 5 MP mobile phone camera to capture wrist vein patterns. This approach has the advantage of not requiring any specialist equipment. A database of 34 individual wrists consisting of three images per wrist was collected. When classified with the radial basis function (RBF) classifier, the dataset achieves a promising RMS error rate of 5.37%. However, whilst the database is not publicly available, the data were also collected over multiple sessions without mentioning if the environmental, lighting and participant factors were consistent. Some modern smartphones already include NIR cameras and LEDs, primarily for facial recognition. In [17], smartphone cameras were used to capture and classify wrist vein patterns. However, it was found that the NIR LEDs have a wavelength of around 960 nm. This is outside the ideal range for vein recognition of 760 nm to 800 nm as the camera system was designed for facial recognition and not vein recognition. When analysed using the scale-invariant feature transform (SIFT) feature extraction tech [18], the system achieved an 18.72% EER. SIFT is also a rather resource expensive technique which meant the system could process 1–2 FPS overall, which is rather low for real-time processing. Most of these studies have not made the datasets publicly available, making it difficult to replicate and extend the research stated in these studies. This is another indicator of wrist vein recognition research being novel. Table 1 summarises the known databases.

From the surveyed databases mentioned in Table 1, it was observed that the captured vein pattern images are generally not directly suitable for feature extraction. The position of the wrist is inconsistent between images and contains more information than required. Raghavendra et al. use hand pegs in [12] to position the wrist which simplifies the region of interest (ROI) identification process. However, this negates the benefit of a contactless system. Fernández et al. use a fixed ROI size in [22], where the user must place their wrist within the ROI. Achban et al. use a cropping technique described in [23] by selecting sub-pixels from the image to take into a new image. A fast rotation, translation and scale compensation algorithm is proposed by Nikisins et al. in [24]. A k-means++ algorithm is used to select an ROI based off the average of two centroids produced by the clustering algorithm, which accounts for translation of the image. Scale is compensated for by making the ROI *k* times smaller when compared to the original image. Rotation is accounted for during the decision making stage.

Enhancing the image is executed through a combination of traditional image enhancement approaches, such as filters and equalisation. Gaussian filters are used in [17,22,24] to smooth the input images, median filters in [17,22,25] and averaging filters in [17], both used to remove noise. Adaptive histogram equalisation (AHE) is a common technique to improve the contrast between veins and skin used in [26], while local thresholding is used for the same purpose in [20,22]. A modified version of AHE, contrast limited adaptive histogram equalisation (CLAHE), is used in [15,17,25]. Regular AHE often over-amplifies the contrast of uniform regions. CLAHE offsets this by clipping the histogram at level *P* and evenly redistributes the clipped pixels throughout the whole histogram [27]. An example of an input image, enhanced image and preprocessed image is shown in Figure 1.

Once the preprocessed image is obtained, the next stage in the system is feature extraction and finally decision making or matching. Feature extraction is the process of capturing important features from the input image. The majority of the literature surrounding feature extraction in wrist vein recognition utilise texture feature extraction methods, which include statistical, structural, transform-based, model-based, graph-based, learning-based and entropy-based approaches [28]. Image recognition is the process of classifying a set of features to determine if they match the known stored features, giving a genuine match or imposter. In more recent years, neural network-based approaches have seen a larger interest in palm vein biometrics; however, research is still limited in terms of wrist vein.

When compared to traditional feature extraction methods that are reviewed in [6], deep learning compensates for differences in the position, translation and rotation of the wrist in the image. They are also able to encompass all signal-processing stages. Kocer et al. utilised deep learning for feature extraction in [29]. A combination of the YOLO (You Only Look Once) and ResNet50 Convolutional Neural Network (CNN) architectures was used. The system was found to segment the ROIs correctly in all testing images and classify the ROIs with a success rate of 95.26%. However, the network was only tested with a small dataset of 66 individuals.

Kurban et al. use three different types of classifiers in [16], radial basis function (RBF) neural network, multi-layer perceptron (MLP) network and a support vector machine (SVM), the former two being neural networks and the latter a traditional classifier. The SVM classifier produced the highest success rate of 96.07% when tested on a private dataset, with the RBF and MLP neural networks both producing a success rate of 94.11%.

Chen et al. utilised CNN for palm vein recognition in [30]. Although the paper focuses specifically on palm vein recognition, the same concepts apply to wrist vein recognition, especially as the CNN is utilised after the ROI segmentation. The initial system was found to achieve an EER of 0.830% and 0.793% for the tested wavelengths and the EER was 0.167% and 0.119% for each wavelength, respectively, in the new CNN + Gabor system.

U-Net is a CNN designed for biomedical image segmentation as proposed by Ronneberger et al. in [31]. The network consists of two symmetric paths: a contracting path designed to capture context and an expansive path to enable localisation. Both paths consist of 3 × 3 convolutional blocks, ReLU activation functions and 2 × 2 max pooling blocks. In one of our previous works, we applied U-Net to palm vein image for segmentation in [32]. The U-Net architecture was modified to decrease the filters in the first and final 2 × 2 convolutional blocks and a Gabor filter was introduced to the first convolutional block to help with extracting the palm vein features. Similar to wrist vein, palm vein suffers from a lack of public data, especially annotated data, which are crucial for the training of neural networks. The training was completed using the HK PolyU palm vein dataset with manually annotated ground truth images. The network achieved a Dice Coefficient of 0.69, which measures the overlap between two binary images, where 1 is fully overlapped and 0 is no overlap.

We developed a Siamese Neural Network for matching palm vein patterns in [10]. A Siamese Neural Network is a network that contains two or more identical sub-networks which process two different pieces of data but share the same weights internally. The outputs of the sub-networks can then be compared using a distance metric to form a probability that two images match. In this paper, a CNN is used as the sub-network and is used to extract a 1D feature set of length 128 from palm vein images. The distance between the two calculated feature sets is then computed using binary cross-entropy. The network was trained using the HK PolyU palm vein dataset which was fed directly into the network. The Adam optimiser was used with a learning rate of 0.0001. Contrastive loss is used as the loss function. Contrastive loss is used in binary classification networks, such as siamese networks and aims to maximise the distance between non-matching feature sets while minimising the distance between matching feature sets. The network achieved an accuracy of 90.5% when sampling each image in the training set five times.

Conducting this review helped gain insights to reach the design decisions for the various sub-systems designed and the integration of the end-to-end wrist vein system.

## 3. Proposed System

This section describes the approach taken to design a low cost complete contactless end-to-end wrist vein biometric recognition system. This system is made up of multiple subsystems as shown in Figure 2. The first stage is to develop the image acquisition device. This device will then be used to acquire the wrist vein image. The second subsection discusses the preprocessing techniques used to transform a wrist vein image and make it suitable for image segmentation, which was done using CLAHE. The third step is to segment the preprocessed images using the U-Net CNN architecture and extract the vein patterns to represent them numerically. In the fourth step, CNN and Siamese Neural Networks were used for image matching. This is one of the most important steps of our recognition system. Finally, a GUI was designed and the subsystems were integrated to achieve a fully working system.

### 3.1. Image Acquisition Subsystem

The image acquisition subsystem comprises NIR LEDs, a NoIR camera with distance mapping calibration. NIR LEDs used in this system are summarised in Table 2 along with their key specifications. The Raspberry Pi Camera Module v2 NoIR comprised of a Sony IMX219 8-megapixel CMOS sensor was chosen to capture the vein patterns. The camera module is small at 25 mm × 23 mm × 9 mm and costs less than USD 100. It was chosen due to these factors and its widespread availability and integration with the Raspberry Pi, a common hobbyist single-board computer. The camera can be controlled via Python by utilising the open-source picamera package. This package allows the user to configure the camera settings, preview the output and capture images from the camera. This influenced the camera choice decision as the previous work carried out by us for deep learning algorithm development was in Python Version 3.8.

### 3.2. Preprocessing Subsystem

Preprocessing is the process of applying transformations to raw input images to make them suitable for image segmentation. As the image acquisition device was being developed in parallel to the deep learning algorithms, it was not possible to test the algorithms with collected data during prototype building. In the interim, the FYO dataset, collected by Toygar et al. in [9], was used for testing these algorithms. The dataset consists of 640 wrist vein images with a resolution of 800 × 600. The dataset is comprised of images collected from volunteers between the ages of 17 and 63, with a gender ratio of 69.27% and 30.73% representing 111 males to 49 females, respectively. The volunteers are mainly from North Cyprus, Turkey, Nigeria, Iran and other parts of the Middle East and Africa to ensure consistent experimentation [9]. The only transformation applied to the FYO dataset images was resizing to a resolution of 256 × 256. In terms of a tradeoff, resizing was found to have been more relevant with respect to performing as opposed to tuning segmentation quality and time. Higher resolutions maintain more detail but take longer to process.

Wrist vein images captured during experimentation lacked contrast between the vein and skin. To improve this, CLAHE was applied as a preprocessing step. This process involves performing traditional histogram equalisation on small tiles throughout the image and then clipping the contrast at a given limit. In this case, CLAHE is applied on tiles with a size of 8 × 8 and clips the contrast with a threshold of 6.

### 3.3. Image Segmentation and Feature Extraction Subsystem

Image segmentation is the process of extracting the vein patterns from the captured images and representing them numerically. The U-Net CNN was used for image segmentation in this paper. The U-Net architecture was chosen due to its success in segmenting palm vein images in our previous work [32].

Significant amounts of labelled training data are also required to perform supervised learning. As the FYO dataset is unlabelled, this would have required days of manual segmentation of the images. Unsupervised learning, which does not require labelled training data, was also considered but dismissed as this method of learning is usually reserved for problems such as clustering and association of input data, not image segmentation. To solve this, signal processing methods proposed in [32] were used to generate masks of the FYO database images. These mask images are then used to train the U-Net. U-net was chosen because the mask image generated by U-net has less noise/artifacts that affect the performance of the matching engine. Using U-Net-generated mask images was preferred because the images were light and computationally efficient to provide as input to the sub-networks of the Siamese Neural Network discussed in Section 3.4.2.

#### 3.3.1. U-Net Architecture for Vein Segmentation

The U-Net CNN architecture consists of two symmetric paths: a contracting path designed to capture context and an expansive path to enable localisation. The architecture is shown in Figure 3.

The contracting path consists of four contracting blocks, each of which contains two 2D convolutions with a 3 × 3 kernel. The number of features doubles in every contracting block. The ReLU activation function is used with the 2D convolutions. These convolutions are followed by max pooling with a 2 × 2 pool size to downsample the feature map, which doubles the number of filters and halves both dimensions of the image. In this U-Net implementation, the input image contains one filter layer which is increased to 16 in the first contracting block. The number of filters then doubles every contracting block up to 128 filters in the final block.

The contracting and expansive paths are joined by two 2D convolutions, again both using a 3 × 3 kernel with a ReLU activation. Similar to the contracting path, the expansive path is built up of blocks. Each block consists of a 2D deconvolution (also known as an up convolution) with a 3 × 3 kernel and 2 × 2 strides. The output is then concatenated with the output of the corresponding block in the contracting path. This is followed by a dropout layer, which at random sets values to zero to prevent overfitting, at a rate of 10%. A 2D convolution with a 3 × 3 kernel size is then performed and is followed with a ReLU activation function, similar to the contracting path. In the original U-Net architecture, the dropout layer is not present and a second convolution layer is present in its place. The dropout layer is inserted to combat the effects of overfitting. The output of the final expansive block is followed by a 1 × 1 convolution to reduce the 16 features down to 1 feature to create a mask image. The sigmoid activation function is used for binary classification, such as in this case, where 1 represents vein and 0 represents not vein. In total, the model consists of 23 tunable layers and 1,179,121 trainable parameters.

The input and output from the network have the same dimensions—(W, H, 1), where *W* is the width of the image and *H* is the height of the image. In this case, the input images have a resolution of 256 × 256. As the U-Net does not contain any fully connected layers, the number of trainable parameters does not increase as the input and output size changes. The U-Net input and output sizes can also change without retraining the network as the weight parameters do not change.

This architecture was proposed in our previous work in [32] and is modified from the original U-Net proposed in [31]. The primary differences include: initial features in the first contracting block decreased from 64 to 16, replacement of one 2D convolution layer with a dropout layer in each expansive block to reduce overfitting, only one output feature in the final output layer and the introduction of a custom Gabor filter kernel in the first contracting block. The input resolution of the U-Net was originally 572 × 527 and was changed to 128 × 128 due to the palm vein input resolution. A resolution of 256 × 256 was used due to the size of the images in the FYO dataset for wrist. Increasing the resolution of the input images allows more information to be passed into the U-Net.

#### 3.3.2. Mask Image Generation Algorithm

To generate labelled data for training the U-Net, a mask image generation algorithm was used as proposed in our previous work in [32]. The image is first denoised by applying a non-local means denoising algorithm and performing opening, which is the process of eroding away the edges of pixel groups and then dilating the remaining pixels with the aim of removing noise. Histogram equalisation is then performed to improve the contrast between vein and skin. An iterative process of erosion and dilation is then performed until no more pixels are removed. A vein pattern image from the FYO database, its generated mask image and the output of the U-Net segmentation algorithm for wrist are shown in Figure 4.

#### 3.3.3. Dice Coefficient

The Dice Coefficient measures how well two binary mask images overlap. The value returned from the Dice Coefficient is between 0, which represents no overlap, and 1, which represents complete overlap. The Dice Coefficient can be calculated as
(1)$$DC=\frac{2|P\cap Y|}{\left|P|+|Y\right|}$$
where *P* is the predicted binary mask and *Y* is the known true binary mask.

#### 3.3.4. U-Net Training

The U-Net was implemented using TensorFlow and Keras 2.1. It was trained on the FYO Wrist Vein database, utilising the region-of-interest images which consists of 640 wrist vein images [9]. The images were divided 80:20 between training and testing. The mask generation algorithm was used to generate known labels for the database. The Adam optimiser was used starting with a learning rate of LR=0.0015 and the Dice Coefficient was used as the loss function. The learning rate of the optimiser was multiplied by 10 after 10 epochs, if no improvement in the loss of the network was seen. This helps the network to continue training after a plateau in loss. The network was trained for 100 epochs, utilising a batch size of 16 images. The result of segmentation via U-Net can be seen in Figure 4.

### 3.4. Image Matching and Decision Making Subsystem

Image matching is the process of comparing two mask images, which are obtained as the result of the image segmentation stage and deciding whether the two images match. Two neural networks were considered for matching: a CNN and a Siamese Neural Network.

#### 3.4.1. Convolutional Neural Network

This paper uses a CNN architecture to compare the mask images and output a probability that the mask images match. The network architecture is shown in Figure 5.

The network consists of convolutional blocks. Each block starts with a 2D convolution layer with a 3 × 3 kernel followed by a ReLU activation function. This is followed by a max pooling layer with a 2 × 2 pool size which halves the input dimensions and doubles the output features. A dropout layer follows this, which at random sets values to 0 to prevent overfitting at a rate of 25%. The output features of the block double every block, starting with 32 features in the blocks immediately after the input images. The network has two input images which follow two identical pathways consisting of two convolutional blocks as described. These blocks are chained in series. Each of these pathways consists of two tunable layers for a total of 18,816 trainable parameters.

After these blocks, the two image feature sets are concatenated and proceed through another two convolutional blocks identical in all aspects except output features. The feature set is then flattened into a 1D tensor with a length of 32,768. A fully connected layer reduces these features down to 512, which is followed by another 25% dropout layer and finally the last fully connected layer, which outputs one value between 0 and 1. All fully connected layers use the ReLU activation function except for the last layer, which uses the sigmoid activation function due to its binary output. In total, the model consists of eight tunable layers for a total of 18,291,201 trainable parameters.

#### 3.4.2. Siamese Neural Network

Siamese Neural Networks are networks that consist of two or more sub-networks and have seen success for vein biometrics in [10]. Siamese networks have the ability to train using 1 or few samples and perform image matching with reasonable accuracy when compared with CNN. A Siamese Neural Network that utilises two identical sub-networks, each of which is provided with one wrist vein binary mask image as created by the U-Net and processes the same, is proposed in this article based on observations from [10]. The high-level network architecture is shown in Figure 6. Two binary mask images are presented to the network—one representing the input image and one representing an enrolled image. Both binary images are processed via two identical neural networks that share the same weights, which produce feature vectors. The Euclidean distance between the two pairs is then calculated and is fed into a sigmoid activation function. If the output of the sigmoid activation function is greater than a preset threshold, the two input images are considered a match.

Figure 6 uses a sub-network. The architecture of this sub-network that processes each input image is shown in Figure 7. The network takes a (W,H,1) binary mask image as an input and produces a vector of length 128, which represents the vein features in the binary mask image. The network consists of three convolutional blocks and a fully connected block. Each convolutional block consists of a 2D convolution layer with a 3 × 3 kernel producing 64 filter layers and is used to extract the features from the image. This is followed by a batch normalisation layer which smoothes the training of the network to increase the speed and stability of training [33]. A batch normalisation layer is also inserted at the very beginning of the network. A ReLU activation function is placed after the batch normalisation layer, followed by an average pooling layer with a 2 × 2 pool size to downsample the feature maps and finally a dropout layer to prevent overfitting of the network.

The output of the convolution blocks is then flattened and run through another batch normalisation layer. The output is then mapped onto a fully connected layer consisting of 128 neurons, which represent the output feature set. The ReLU activation function is used in the dense layer. In total, the model consists of 11 tunable layers with 7,678,602 trainable parameters.

Contrastive loss [34] is commonly used as the loss function for siamese networks. The goal of contrastive loss is to maximise the distance between non-matching feature sets and minimise the distance between matching feature sets. Contrastive loss can be calculated using
(2)$$L=mean(\left(1-Y\right)\left(P^{2}\right)+Y\left(max{\left(M-P,0\right)}^{2}\right))$$
where *L* is the calculated loss, *Y* is the known truth values, *P* is the predicted truth values and *M* is the baseline distance for pairs to be considered dissimilar. The margin is commonly set to 1, which is the value used throughout this paper.

#### 3.4.3. Network Training

Both networks were implemented using TensorFlow and Keras 2.1 and were trained using the FYO wrist vein dataset. The dataset was split 80:20 between training and testing for a total of 256 wrists for training and 64 wrists for testing and validation. It is important to note that each wrist was captured twice between two imaging sessions. The U-Net discussed in Section 3.3.1 was used to generate mask images for all images.

As matching is the process of comparing two images, a dataset of image pairs was constructed. Each wrist in the FYO dataset had two images across two sessions. These two images were paired and were assigned a “true” label. The image from the first session is then paired with a random image from the second session that does not match and is assigned a “false” label. This produces a wrist pair dataset of 512 pairs for training and 128 pairs for testing and validation.

To attempt to teach the network to compensate for translation, rotation and scale of the vein patterns, a set of augmentations were applied to the images before training or testing. These include rotating the image $\pm 15^{{}^{\circ}}$, zooming the image between 40% and 60% and translating the image ±10% in both vertical and horizontal directions. These augmentations, when applied, were re-done for every epoch so that the network was unlikely to see the same vein pattern in the same position.

Both networks were trained using the Adam optimiser with a learning rate of LR=0.0003. The CNN utilised the binary cross entropy loss function and a batch size of 32, while the Siamese Neural Network utilised contrastive loss with a batch size of 16.

### 3.5. Graphical User Interface

A graphical user interface (GUI) was developed to allow users to capture wrist vein images, segment them using U-Net and compare them using various matching networks. As the camera in use is the Raspberry Pi NoIR camera, the application needed to run on the Raspberry Pi itself. Python was chosen as the language of choice due to the machine learning algorithms also being developed in Python. PyQt5, a Python interface to the Qt GUI library, was used to design the application.

Due to the Raspberry Pi having significantly less computation resources as compared to a desktop computer, it is not possible to install and run TensorFlow 2 without significant work-arounds. TensorFlow Lite (TFLite) is a lightweight version of TensorFlow that has the ability to run inference on small embedded platforms such as Raspberry Pi. Regular TensorFlow models can be compiled into a TensorFlow Lite model. The GUI application uses a TFLite U-Net model to segment the wrist vein images. To support the future development of matching networks, the application can also load different TFLite models from a folder to use in the matching process. The CNN and siamese neural network models are pre-loaded into the initial setup. The segmentation and matching processes of the GUI are shown in Figure 8 and Figure 9, respectively.

## 4. Results and Discussion

### 4.1. Image Acquisition

The image acquisition device was produced and evaluated. Of the four NIR wavelengths proposed, the 860 nm LEDs provided the clearest wrist vein images. This is likely due to the 860 nm LEDs having a high radiant intensity. The other wavelengths would require increasing the shutter speed and/or ISO of the camera, which in turn would introduce more noise into the wrist vein image. The final image acquisition device is shown in Figure 10 with the 860 nm LEDs. The acquisition system implementation is intended to be filed as a patent.

### 4.2. Image Segmentation

The U-Net used for image segmentation was trained and tested using the FYO wrist vein dataset. The dataset of 640 images was split 80:20 between training and testing. The network was trained for 107 epochs as training was stopped early due to the plateau of the accuracy metric. In this case, the accuracy is binary accuracy and is calculated based on how many pixels the model correctly predicts. The Dice Coefficient is another metric which measures the overlap of the predicted and true mask images as described in Section 3.3.3. The model achieved a binary accuracy of 90.5% and a Dice Coefficient of 0.723. Figure 11 shows the Dice Coefficient and binary accuracy for one training session.

Overall, the U-Net is successful at segmenting wrist vein images. Compared to the results of palm vein segmentation using U-Net from [32], which observed a best Dice Coefficient of 0.697, the U-Net performs on par or better when processing wrist vein images. This is likely attributed to the increase in input resolution from 128 × 128 to 256 × 256, which maintains more detail in the vein patterns. Another factor to consider is the lack of palm patterns on the wrist which provides a clearer region of interest to capture the vein patterns.

### 4.3. Image Matching

Both image matching neural networks were evaluated using the FYO wrist vein dataset. The networks were provided wrist vein mask images segmented using the discussed U-Net architecture. Binary accuracy and F1-score are used to evaluate the matching models. Binary accuracy is the percentage of pairs the model predicted correctly, while the F1-score is the harmonic mean of the precision and recall metrics. Here, 100% indicates a perfect model and 0% represents the opposite. These are the metrics used to evaluate the model. The F1-score can be calculated with
(3)$$F=\frac{2\cdot TP}{2\cdot TP+FP+FN}$$
where TP represents the number of true positives, FP represents the number of false positives and FN represents the number of false negatives.

#### 4.3.1. Convolutional Neural Network

The CNN was trained for 200 epochs. The binary cross entropy loss function was used for training and validation. Binary accuracy and F1-score were used as the accuracy metrics. The network achieved a binary accuracy of 65.6% and an F1-score of 73.3% after training for 200 epochs. Figure 12 and Figure 13 show the evaluation metrics of the network for the training and validation datasets for one training session.

Likely due to the size of the network, with almost 70,000,000 trainable parameters, the network learns at a slow rate as evidenced by the number of epochs required to increase the validation accuracy reported as compared to the siamese and U-Net neural networks. Even after 200 epochs the loss is still trending downwards, indicating the network still has room to learn. However, as can be seen in the plots in Figure 12 and Figure 13, the validation accuracy and F1-score plateaus early at approximately 50 epochs while the training metrics continue to increase, which may indicate that the network is failing to learn and is instead overfitting to the training data. This may be attributed to the relatively small dataset, as in general, larger networks require more data to train. Augmentation of the input images was performed to help this by effectively increasing the size of the dataset but does not help the fact there are only 256 unique pairs for testing the network. This small number of pairs is also likely why the validation F1-score is consistently higher than the training F1-score, as there is a chance the training dataset was allocated higher quality images that are easier to match.

#### 4.3.2. Siamese Neural Network

The Siamese Neural Network was trained for 100 epochs. Contrastive loss was used for training and validation. Similar to the CNN, binary accuracy and F1-score were used as the accuracy metrics. The network achieved a binary accuracy of 85.1% and an F1-score of 84.7% after training for 100 epochs. Figure 14 and Figure 15 show the contrastive loss, binary accuracy and F1-score of the network for the training and validation datasets for one training session.

Compared to the CNN, the Siamese Neural Network provides a much better accuracy and F1-score even after only training for half the number of epochs. The loss of the network has begun to plateau at 100 epochs, indicating the network has reached a minima. If the network has reached a local minima, the accuracy and F1-score may be able to further be increased by adjusting the learning rate, which was performed during U-Net training.

## 5. Conclusions and Future Work

In this paper, a small and relatively low-cost wrist vein image acquisition device was designed to capture high-quality images of wrist vein patterns. It was found that NIR light with a wavelength of 860 nm provides the highest quality vein pattern images; however, further evaluation is required to determine whether the radiant intensity of the LEDs influenced this result. The U-Net neural network architecture has been applied to the problem of image segmentation to extract the vein pattern features from the input images, achieving a Dice Coefficient of 0.723 when tested on the FYO wrist vein dataset. Convolutional and Siamese Neural Networks have been applied to the problem of image matching, with the Siamese Neural Network showing promising results, achieving an F1-score of 84.7% when tested on the FYO wrist vein dataset. To combine all aspects of the research together, a GUI was developed to allow users to capture wrist vein images, segment them with U-Net and compare them with locally saved images using their own matching neural networks or one of the provided networks.

Due to supply constraints, it was not possible to obtain LEDs with similar radiant intensities for each wavelength. Due to this, the LED chosen for 860 nm had a radiant intensity over 200× more than the next largest LED, as can be seen in Table 2. This has likely skewed the image acquisition results towards 860 nm as more light allows us to reduce the shutter speed and ISO of the camera to allow less light and less noise to enter the camera. Current limiting resistors were added to the design to ensure safety measures were in place with the LED usage. It would be ideal to source LEDs with similar radiant intensities as an extension of this research to provide more validation that 860 nm is absorbed optimally.

Although the Siamese Neural Network has provided promising results for image matching, it can be optimised further by tuning the hyperparameters. The GUI has been designed to allow users to easily test their image-matching algorithms in real-time settings with an image acquisition system. Users can simply compile their TensorFlow models to TensorFlow Lite models, upload the model to the Raspberry Pi and select the model in the GUI. The only requirement is that the model take two mask images as inputs and output the probability that the two provided images match. The next step could be to further investigate the image matching algorithms and evaluate them using the end-to-end system presented in this paper.

## Figures and Tables

**Figure 1 sensors-23-03132-f001:**
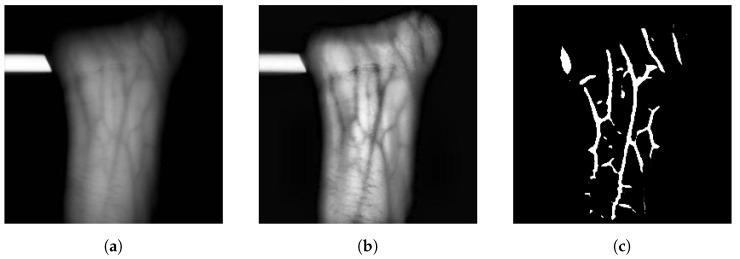
Wrist Vein Example Images. (**a**) input, (**b**) Enhanced Vein Image, (**c**) Vein Features.

**Figure 2 sensors-23-03132-f002:**
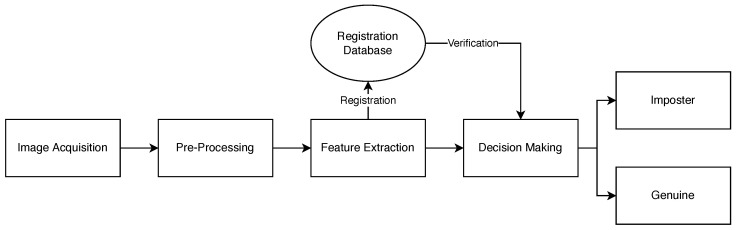
Wrist Vein Recognition System Flowchart.

**Figure 3 sensors-23-03132-f003:**
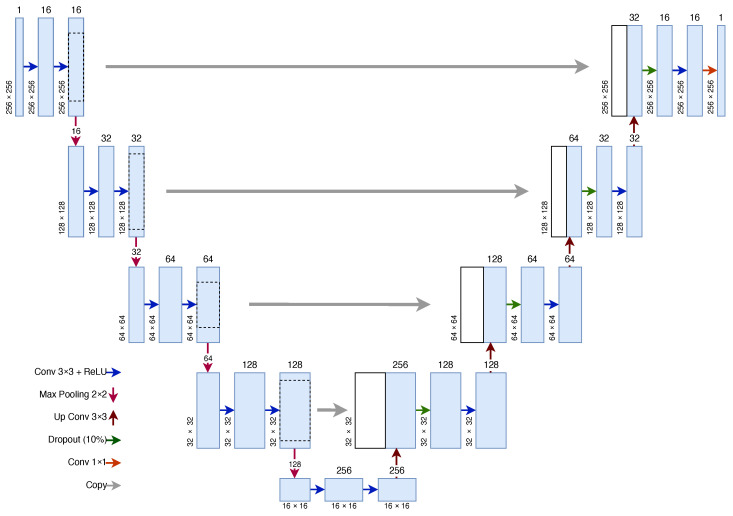
U-Net Architecture.

**Figure 4 sensors-23-03132-f004:**
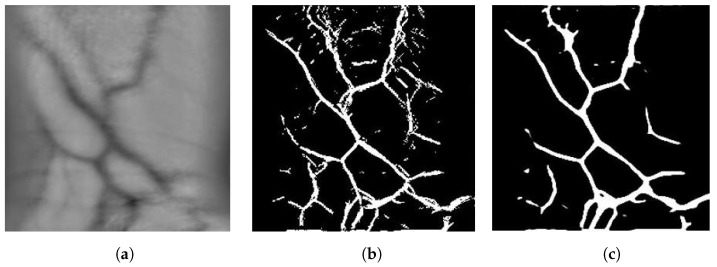
Generated mask images for the original image from FYO. (**a**) Original Image, (**b**) Mask Generated, (**c**) Mask Generated with U-Net.

**Figure 5 sensors-23-03132-f005:**
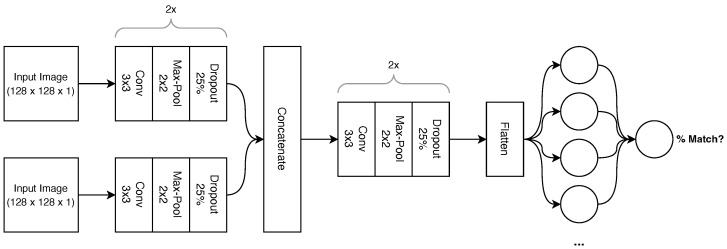
CNN Matching Network Architecture.

**Figure 6 sensors-23-03132-f006:**
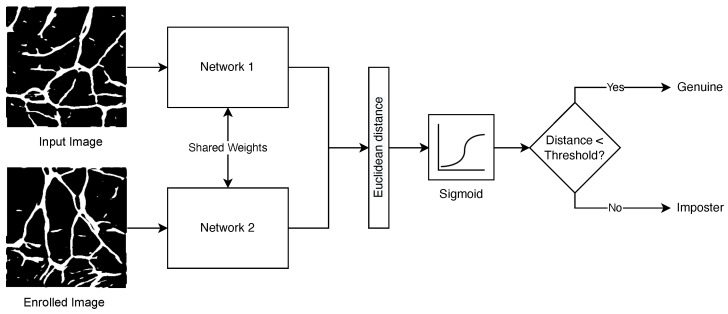
Siamese Neural Network Architecture with sub-network.

**Figure 7 sensors-23-03132-f007:**
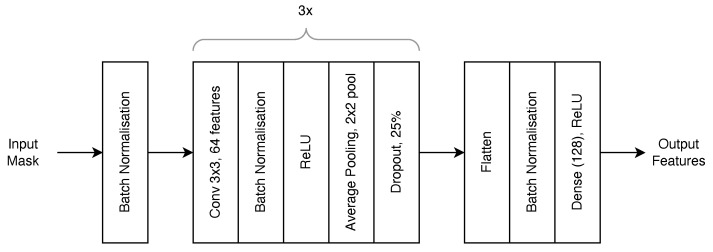
Sub-network Siamese Feature Network Architecture.

**Figure 8 sensors-23-03132-f008:**
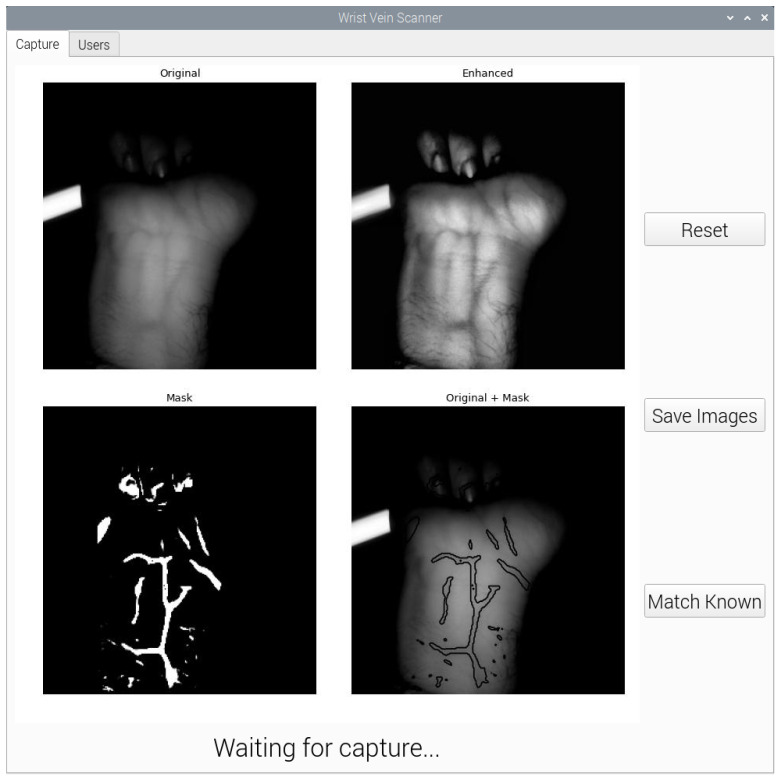
Wrist Vein GUI Segmentation Process.

**Figure 9 sensors-23-03132-f009:**
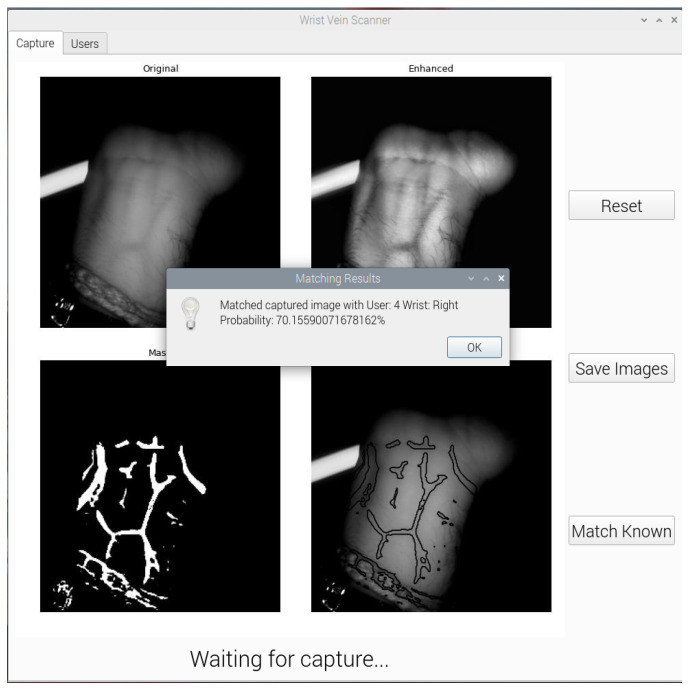
Wrist Vein GUI Matching Process.

**Figure 10 sensors-23-03132-f010:**
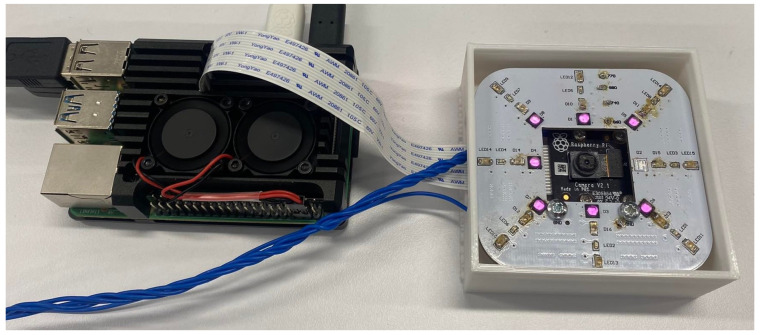
A Developed Filed to Patent Image Acquisition Device.

**Figure 11 sensors-23-03132-f011:**
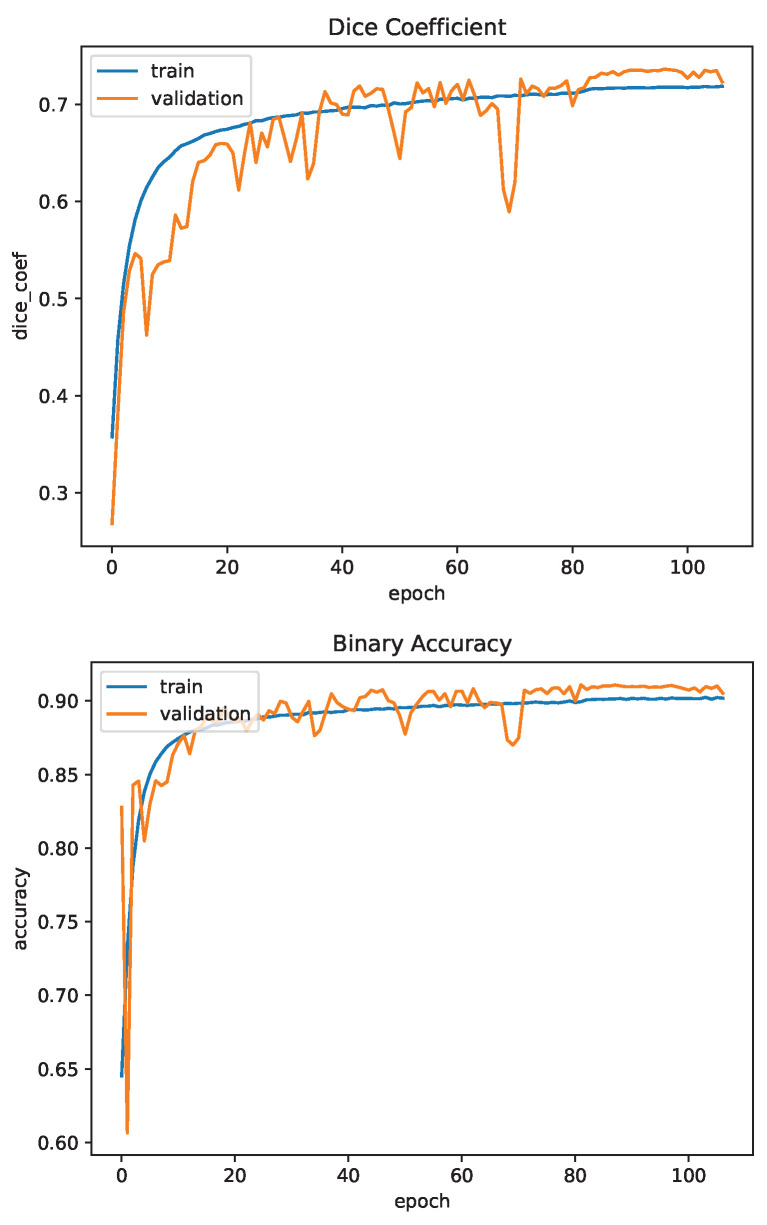
U-Net Training Evaluation.

**Figure 12 sensors-23-03132-f012:**
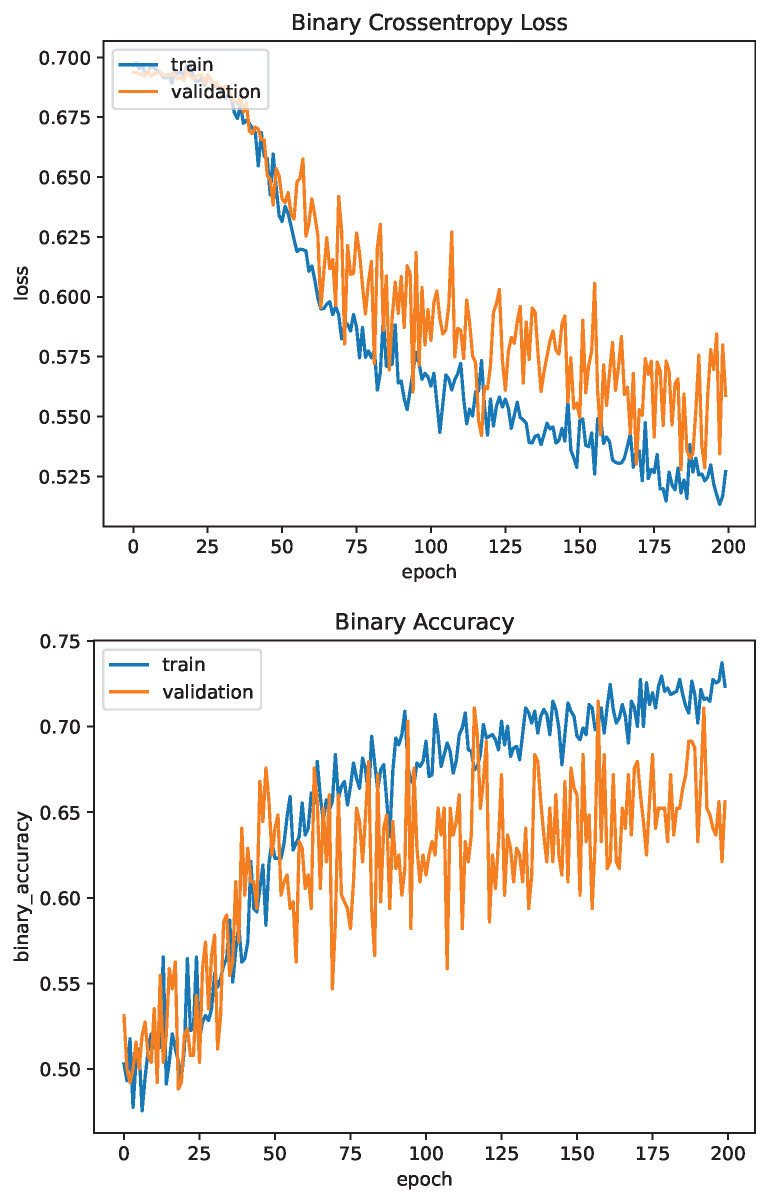
Convolutional Network Training Evaluation: Loss and Accuracy.

**Figure 13 sensors-23-03132-f013:**
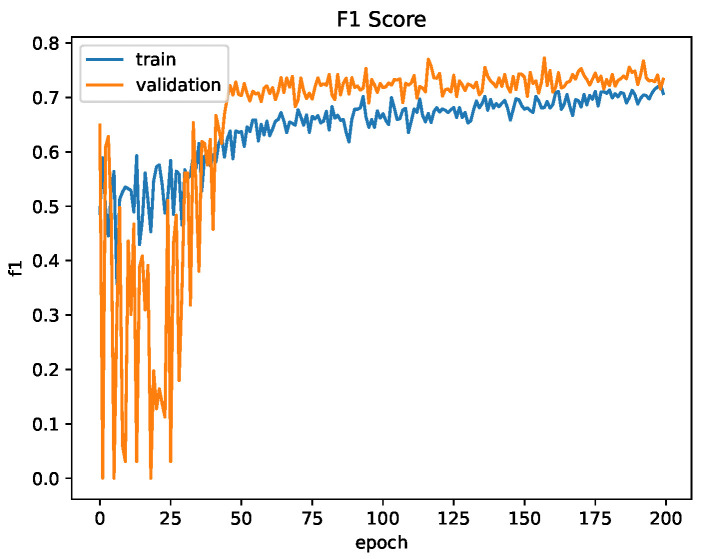
Convolutional Network Training Evaluation: F1-Score.

**Figure 14 sensors-23-03132-f014:**
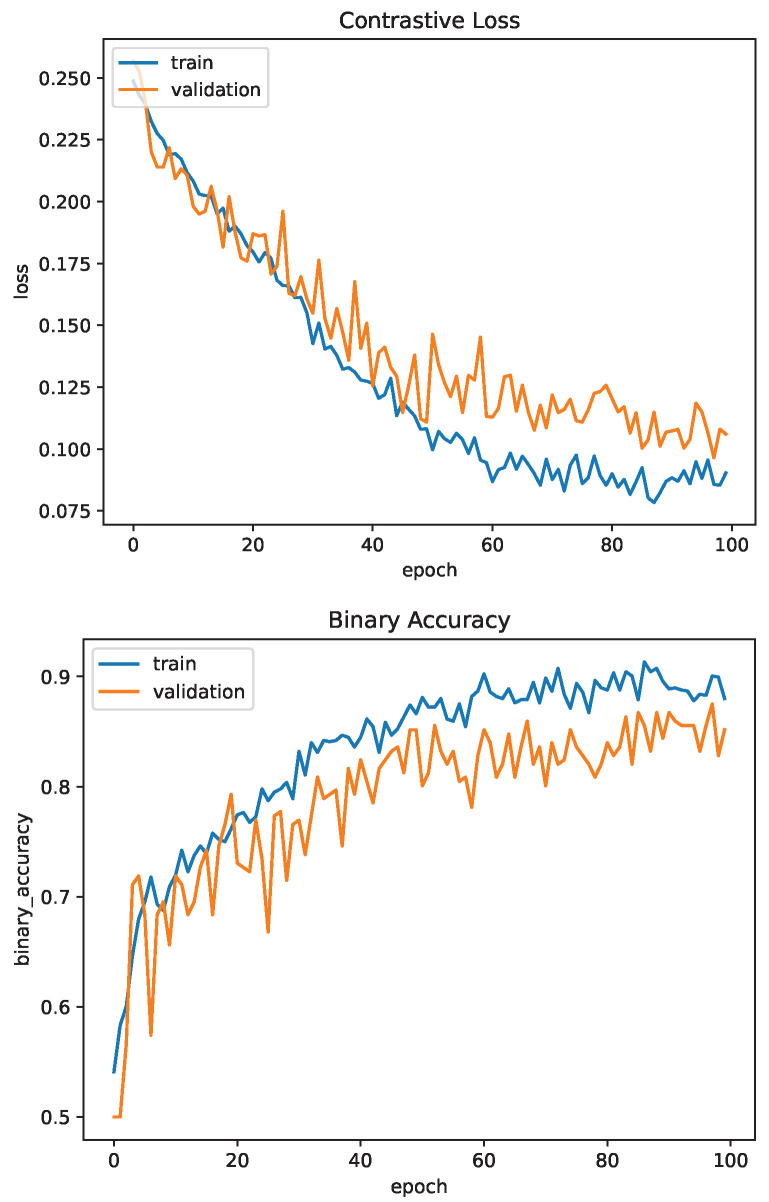
Siamese Network Training Evaluation: Loss and Accuracy.

**Figure 15 sensors-23-03132-f015:**
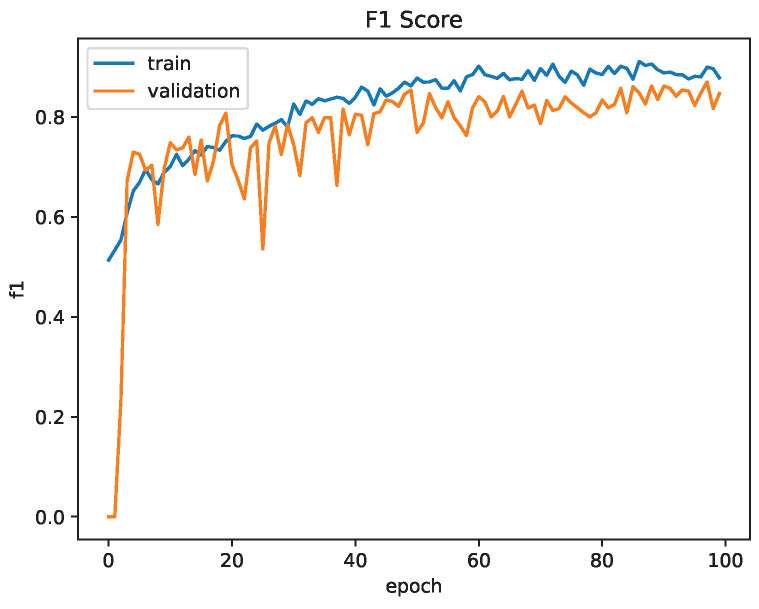
Siamese Network Training Evaluation: F1-Score.

**Table 1 sensors-23-03132-t001:** Wrist Vein Pattern Databases.

Name	Participants	Wrist	Samples	Sessions	Total	Camera	NIR
PUT [19]	50	2	4	3	1200	Unknown	Unknown
Singapore [20]	150	2	3	Unknown	900	Hitachi KP-F2A	850 nm
FYO [9]	160	2	2	1 ^1^	640	1/3 inch infrared CMOS	Unknown
UC3M [21]	121	1	5	Unknown	605	DM 21BU054	880 nm
UC3M-CV1 [15]	50	2	6	2	1200	Logitech HD Webcam C525	850 nm
UC3M-CV2 [17]	50	2	6	2	1200 per device	Xiaomi Pocophone F1, Xiaomi Mi 8	960 nm
Kurban et al. [16]	17	2	3	Unknown	102	5MP mobile phone	No NIR
Pascual et al. [14]	30	2	6	Unknown	360	DM 21BU054	880 nm
Fernández et al. [22]	30	Right	4	1	120	CCD Camera	880 nm

^1^ The FYO database was collected over two sessions separated by only 10 min.

**Table 2 sensors-23-03132-t002:** Specification of Near-Infrared Light Used In The Image Acquisition Device.

Wavelength	Reasoning	Model	Forward Voltage	Forward Current	Radiant Intensity
740 nm	Successfully used in wrist vein literature	OIS 330 740 X T	1.7 V	30 mA	6 mW/sr
770 nm	Absorbed best by deoxygenated hemoglobin	OIS 330 770	1.65 V	50 mA	6 mW/sr
860 nm	Successfully used in wrist vein literature	SFH 4715AS	2.9 V	1 A	1120 mW/sr
880 nm	Successfully used in wrist vein literature	APT1608SF4C-PRV	1.3 V	20 mA	0.8 mW/sr

## Data Availability

Not applicable.

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
