# Peer review of "Deep Learning-Based Wrist Vascular Biometric Recognition"

_sensors, 2023, doi:10.3390/s23063132_

Round 1

Reviewer 1 Report

General

I would like to congratulate the Authors on their successful and innovative work. This article showcases the methods and results very well.

The following comments are mainly targeting the clarity, conciseness, and flow of the article, alongside a few requests for further details.

I think with minor modification/refinement, this will be an excellent article in a novel area of research. I look forward to seeing how this work progresses.

English and grammar

No significant errors, but at times the text lacks conciseness and the grammar can make the text unnecessarily cumbersome. For example, "Wrist vein recognition has seen less research when compared to its palm and finger vein counterparts, however, it is of interest due to the thickness of the skin around the wrist being thin, making the wrist vein more accessible in the image acquisition stage of the biometric system [7], clarity of image captured due to less noise from external information such as lack of palm lines and fingerprints, and the size of the region of interest being small since it is the wrist." This single sentence starts on line 55 and finishes on line 60. There are several sentences like this which need breaking up to aid the reader, alongside a general tidy up of unnecessary text to aid conciseness.

Also to this end, there are lots of cases of repetition. As a specific example, the information given on line 463 is almost immediately stated again on line 471. This occurs frequently throughout the article. Obviously this is sometimes required to set the context of a new discussion point, but the frequency here seems excessive. 

Text in Figures 2, 3, 5, 6, 7, 11, 12, 13, 14, and 15 could be made slightly larger.

Section 1 - Introduction

Content overall good. Gives a good introduction to the topic and the motivation to the work. The mention of Covid-19 seems strange though - almost irrelevant. The authors have positioned the pandemic as a significant motivation for their work by prominently mentioning it in the first sentence of the abstract; one would think that a bigger justification would have been provided, beyond the short statement on line 33. Makes the use of Covid-19 in the abstract seem like a hook rather than a genuine point of motivation.

In the paragraph starting on line 61, the authors draw the distinction between authentication and identification, but then make no reference to these again. Why is this information relevant to this work? Have you targeted a specific application of the two? Is one more difficult to achieve than the other? This is a case of context without a point.

Section 2 - Literature review

This section gives a well rounded overview of the area.

Up until the paragraph beginning on line 178, the authors had mentioned "deep learning", then switch to "deep neural networks", then switch back. Was this intentional? These two topics are very related, but different; the former being more of a general term than the latter. There is no comment on how the two approaches compare in this application.

Most of the technical terms introduced are described and referenced well. A couple could be improved, e.g. SIFT and EER have no accompanying references.

When describing the IR illumination the phrase "X of NIR LEDs and an illumination wavelength of Ynm" is consistently used. This is cumbersome and unnecessarily complicates the point the author is making. This goes back to the above point on being concise - "X Ynm LEDs" provides the same information with much less text.

Section 3 - Proposed system

Line 240, what is the relevance of the colour image acquisition?

Line 242, could this value also be given in another currency (e.g. USD?) to make this more relatable to an international reader?

End of 3.1, the mention of the dataset needs moving to the relevant sub-section.

Table 2, what is the acronym SR under radiant intensity? Should this be sr, for steradian? Also the authors do not comment on the powers of the LEDs, especially the 860nm 1.12W; were there any safety measures or requirements put in place? Caption is also insufficient, in that it doesn't describe the table contents.

Sub-section 3.2, why is this sort of pre-processing required for CNNs in the first place? CNNs can usually operate with quite complex images. Doesn't this pre-processing procedure remove potentially useful information from the images?

Could the authors comment on any demographic biases within the FYO dataset? To ensure this system works consistently, the FYO dataset would need to be a representative sample of the population, for example different ages and ethnicities.

Figure 4, caption needs expanding. The figure should work independent of the text. The sub-figures should also be annotated.

Line 336/337, "reduced by a factor of 0.1" is confusing. Wouldn't this mean multiplying by 10?

Section 3.3, the authors have not mentioned why they are using the U-Net model over their mask image generation algorithm. If the latter has been used to train the former, then why is it needed at all if the algorithm already works? What are the benefits of using U-Net? According to Ref 29, they perform very similarly; which isn't surprising given the training method.

Line 344, CNN acronym has already been defined, but hasn't been used here. 

Line 360, final dropout layer is missing from Figure 5.

Figure 5, caption is insufficient and doesn't mention CNN.

Figures 6/7, it should be made clearer how these two structures are related within the diagrams.

Section 3.4.2, it is not clear whether this architecture has been taken from Ref 10, a modified version of Ref 10, or the Authors own interpretation of the architecture used in Ref 10.

Line 396/397, could a reference be provided for this point?

End of Section 3.4.3, why was this augmentation conducted on an epoch basis? Could an "effectively" larger dataset not have been generated by this augmentation process prior to training? What would have been the potential issues here?

Section 4 - Results and Discussion

Figures 8/9, there doesn't seem to be much difference between these figures other than the pop-up window, is this right? Could one be removed?

Section 4.3.2, the Authors mention a possible issue with the CNN could have been the size of the dataset relative to the number of trainable parameters. Could the same information be provided for the siamese neural network? Surely a point of discussion would be how the two compare if there are similar, or even more, trainable parameters?

Section 5 - Conclusion & future work

A very nice summary of the work conducted.

As a final point, it would be good to put this work into the context of the wider field, along with potential applications. Improvements required before specific applications could be targeted would also be of interest to the reader, e.g. would the ~85% accuracy provided by the siamese neural network good enough for real world applications?

Author Response

We would like to thank the reviewer for raising important points to improve the paper. The reviewer’s comments are very targeted, highly valuable, and demonstrates the

Reviewer’s extensive knowledge of the subject. We appreciate the valuable inputs and have meticulously incorporated all the relevant changes and clarified the questions. We are sure these inputs resulted in the article becoming more valuable to the reader and wider scientific research community.

Comment

General

I would like to congratulate the Authors on their successful and innovative work. This article showcases the methods and results very well.

The following comments are mainly targeting the clarity, conciseness, and flow of the article, alongside a few requests for further details.

I think with minor modification/refinement, this will be an excellent article in a novel area of research. I look forward to seeing how this work progresses.

Response

We would like to thank the reviewer for the valuable inputs and precise points mentioned that would help improve this research article. We understand and appreciate the time and knowledge invested into it, and it is our pleasure to address all the points mentioned.

Below is the point by point response to the comments:

Comment

English and grammar

No significant errors, but at times the text lacks conciseness and the grammar can make the text unnecessarily cumbersome. For example, "Wrist vein recognition has seen less research when compared to its palm and finger vein counterparts, however, it is of interest due to the thickness of the skin around the wrist being thin, making the wrist vein more accessible in the image acquisition stage of the biometric system [7], clarity of image captured due to less noise from external information such as lack of palm lines and fingerprints, and the size of the region of interest being small since it is the wrist." This single sentence starts on line 55 and finishes on line 60. There are several sentences like this which need breaking up to aid the reader, alongside a general tidy up of unnecessary text to aid conciseness.

Response

We acknowledge this issue that has rightly been pointed out by the reviewer. In the revised version, we have reviewed these extensive long statements and broken them down into concise sentences so that the paper is easier to read and is able to convey clear information.

Comment

Also to this end, there are lots of cases of repetition. As a specific example, the information given on line 463 is almost immediately stated again on line 471. This occurs frequently throughout the article. Obviously this is sometimes required to set the context of a new discussion point, but the frequency here seems excessive.

Response

We have reviewed the repetitiveness of information throughout the article and edited it to avoid the same. In line 463, the definition of binary accuracy and F1 score is mentioned. Whereas, in line 471, the reader is pointed to the observed value of the defined parameters with the help of figures 12 and 13. We acknowledge that the choice of words make it seem like it is repeated information and hence, we have now re-worded it to be less confusing and clear.

Comment

Text in Figures 2, 3, 5, 6, 7, 11, 12, 13, 14, and 15 could be made slightly larger.

Response

We acknowledge this comment and have increased the font size and image size in the relevant areas of the article.

Section 1 - Introduction

Content overall good. Gives a good introduction to the topic and the motivation to the work. The mention of Covid-19 seems strange though - almost irrelevant. The authors have positioned the pandemic as a significant motivation for their work by prominently mentioning it in the first sentence of the abstract; one would think that a bigger justification would have been provided, beyond the short statement on line 33. Makes the use of Covid-19 in the abstract seem like a hook rather than a genuine point of motivation.

Response:

The idea of mentioning Covid-19 was primarily to emphasize on the recent demand for contactless biometric systems. Covid-19 is referred to in this article only in this context as the system developed is completely contactless. However, we appreciate the input from the reviewer and understand how this could be unsettling to the reader, especially when it is mentioned in the abstract. Therefore, we removed the reference to Covid-19 in the abstract as it is not directly relevant and retained the general notion of increase in preference for contactless biometrics in the introduction section

Comment:

In the paragraph starting on line 61, the authors draw the distinction between authentication and identification, but then make no reference to these again. Why is this information relevant to this work? Have you targeted a specific application of the two? Is one more difficult to achieve than the other? This is a case of context without a point.

Response:

The difference between authentication and identification is that the former is one-to-one and latter is one-to-N. This article explicitly distinguishes the operational mode (either authentication or identification). This is the reason the operational mode specifications are explicitly introduced in line 61. Identification is difficult to achieve as compared to authentication as it involves a wider search and match between the probe and candidate. The deep learning based wrist vein biometric system developed is capable of operating in both, authentication and identification mode. Wider literature uses these terms interchangeably without appreciating the fundamental difference which is critically decisive of how the biometric system works. This is the exact reason for explicitly mentioning this before going into specific and complex details of the article.

Comment

Section 2 - Literature review

This section gives a well rounded overview of the area.

Up until the paragraph beginning on line 178, the authors had mentioned "deep learning", then switch to "deep neural networks", then switch back. Was this intentional? These two topics are very related, but different; the former being more of a general term than the latter. There is no comment on how the two approaches compare in this application.

Response:

We acknowledge this point raised by the reviewer. This switch was not intentional. The former is more representative of what we intend to discuss. As such this has been carefully reviewed and replaced to “deep learning” for consistency and better readability.

Most of the technical terms introduced are described and referenced well. A couple could be improved, e.g. SIFT and EER have no accompanying references.

Response:

We acknowledge this point raised by the reviewer. The modified version of the paper has been updated with appropriate references including SIFT and EER.

When describing the IR illumination the phrase "X of NIR LEDs and an illumination wavelength of Ynm" is consistently used. This is cumbersome and unnecessarily complicates the point the author is making. This goes back to the above point on being concise - "X Ynm LEDs" provides the same information with much less text.

Response:

We acknowledge this point raised by the reviewer. The modified version of the paper has been updated where the LED information has been changed and is in the format "X Ynm LEDs".

Section 3 - Proposed system

Line 240, what is the relevance of the colour image acquisition?

Response:

We acknowledge this point raised by the reviewer. This information is not of much information and has been removed in the updated manuscript.

Line 242, could this value also be given in another currency (e.g. USD?) to make this more relatable to an international reader?

Response:

We acknowledge this point raised by the reviewer. To make it more relatable to international readers, the currency has been appropriately updated to USD.

End of 3.1, the mention of the dataset needs moving to the relevant sub-section.

Response:

We acknowledge this point raised by the reviewer. This has been moved to the relevant sub-section(i.e. 3.2) in the updated manuscript.

Table 2, what is the acronym SR under radiant intensity? Should this be sr, for steradian? Also the authors do not comment on the powers of the LEDs, especially the 860nm 1.12W; were there any safety measures or requirements put in place? Caption is also insufficient, in that it doesn't describe the table contents.

Response:

We acknowledge this point raised by the reviewer. The acronym SR is for radiant intensity. As rightly pointed out it should be ‘sr’. This has been updated in the current version of the manuscript. A comment on the power of the LEDs has been added where we mention that current limiting resistors were put in place to ensure there is no overcurrent leading to the LED damage. The caption of the table was insufficient and has been appropriately modified in the updated version of the manuscript.

Sub-section 3.2, why is this sort of pre-processing required for CNNs in the first place? CNNs can usually operate with quite complex images. Doesn't this pre-processing procedure remove potentially useful information from the images?

Response:

The pre-processing  step is intended only to enhance the image. During the experimentation stage, we analyzed the results with pre-processing and without pre-processing. Significant improvement in matching accuracy was observed when using pre-processing, this is the reason it is used and we can confirm that no useful information that would aid the matching engine’s performance has been lost.

Could the authors comment on any demographic biases within the FYO dataset? To ensure this system works consistently, the FYO dataset would need to be a representative sample of the population, for example different ages and ethnicities.

Response:

We acknowledge this point raised by the reviewer. The FYO dataset does consider inclusion of volunteers from a broad age group, various ethnicities, and genders to ensure consistency. This is now updated in the current version of the manuscript with relevant referencing.

Figure 4, caption needs expanding. The figure should work independent of the text. The sub-figures should also be annotated.

Response:

We acknowledge this point raised by the reviewer. The captions have been reviewed and updated. Sub-figures are now independently annotated with appropriate captions. 

Line 336/337, "reduced by a factor of 0.1" is confusing. Wouldn't this mean multiplying by 10?

Response:

We acknowledge this point raised by the reviewer. This indeed means multiplying by 10. This has been accordingly updated in the current version of the manuscript.

Section 3.3, the authors have not mentioned why they are using the U-Net model over their mask image generation algorithm. If the latter has been used to train the former, then why is it needed at all if the algorithm already works? What are the benefits of using U-Net? According to Ref 29, they perform very similarly; which isn't surprising given the training method.

Response:

The information about the U-net model and its superiority over the mask image generation algorithm can be seen in Ref 29. U-net was chosen because the mask image generated by U-net has less noise/ artifacts that affect the performance of the matching engine. The mask image generation algorithm can be used with the entire system. This was attempted, however the matching score was not desirable at all. We have now added a comment about this. This clarification would be beneficial to the reader to appreciate the choice of U-net model.

Line 344, CNN acronym has already been defined, but hasn't been used here.

Response:

We acknowledge this point raised by the reviewer and this has been updated to ‘CNN’ in the updated version of the manuscript.

Line 360, final dropout layer is missing from Figure 5.

Response:

“A dropout layer follows this, which at random sets values to 0 to prevent overfitting at a rate of 25%”. We believe this was something that was missed by the reviewer. Figure 5 does have a dropout layer 25% marked on it.

Figure 5, caption is insufficient and doesn't mention CNN.

Response:

We acknowledge this point raised by the reviewer and this has been updated in the current  version of the manuscript.

Figures 6/7, it should be made clearer how these two structures are related within the diagrams.

Response:

We acknowledge this point raised by the reviewer. Indeed figure 7 is the network stage represented in figure 6. We understand how this can be less intuitive. We have added statements that explicitly mention this, and are much easier to relate now. This is addressed in the updated version of the manuscript. 

Section 3.4.2, it is not clear whether this architecture has been taken from Ref 10, a modified version of Ref 10, or the Authors own interpretation of the architecture used in Ref 10.

Response:

We acknowledge this point raised by the reviewer. Ref 10 is from our own work. The architecture used in this article is inspired from Ref 10 and is an optimized version much suited for wrist vein images. We acknowledge this might not be explicitly clear from a reader's point of view. Therefore, we now have added statements which can clarify part making it less confusing.

Line 396/397, could a reference be provided for this point?

Response:

We acknowledge this point raised by the reviewer. A relevant reference has been provided to support this statement about contrastive loss.

End of Section 3.4.3, why was this augmentation conducted on an epoch basis? Could an "effectively" larger dataset not have been generated by this augmentation process prior to training? What would have been the potential issues here?

Response:

When a probe image is being compared with a candidate image, it is more effective to have  augmented versions of the probe image during the training/ testing phase itself within the epoch. This helps the model learn in a realistic way where the variations that could occur in a real testing environment is replicated. This is the reason augmentation was conducted on an epoch basis. A larger database would not suit this as this scenario is quite specific to biometrics.

Section 4 - Results and Discussion

Figures 8/9, there doesn't seem to be much difference between these figures other than the pop-up window, is this right? Could one be removed?

Response:

We believe both images are relevant. Figure 8 shows the GUI with all the features available in it in a more general way, whereas figure 9 is showing a particular case of image matching. In this case, the captured image was matched to the one that was stored in the database and it had a confidence level of 70.15%. Both these images convey different aspects of the developed system and help the reader appreciate the entire working of the system. Therefore, we would like to retain both the images.

Section 4.3.2, the Authors mention a possible issue with the CNN could have been the size of the dataset relative to the number of trainable parameters. Could the same information be provided for the siamese neural network? Surely a point of discussion would be how the two compare if there are similar, or even more, trainable parameters?

Response:

The statements in Section 4.3.2 highlights the issues with CNN based approach and then justify the need/advantage of using Siamese networks. The beauty of Siamese networks is its ability to train using 1 or few samples and perform image matching with practical accuracy. We believe this is not well conveyed in  Section 4.3.2 and hence leading the reader to this question. Therefore we have now added more clear statements to the document that clarify this bit.

Section 5 - Conclusion & future work

A very nice summary of the work conducted.

As a final point, it would be good to put this work into the context of the wider field, along with potential applications. Improvements required before specific applications could be targeted would also be of interest to the reader, e.g. would the ~85% accuracy provided by the siamese neural network good enough for real world applications?

 Response:

This work indeed can be extended to other fields. Most of the literature report higher accuracy rates and other specific performance parameters but do not have any working end-to-end system. This is the first work where a reader can follow the steps in the article and build the entire system. Once the system is built it can always be optimized for accuracy which is higher than 85%. The focus of this article is to report a fully functional end-to-end system using deep learning.

Reviewer 2 Report

In this paper, the authors proposed a biometrics method using wrist vein patterns. The DB used for verification is FYO, the vein extraction accuracy is 0.723 based on the Dice correlation coefficient, and the recognition accuracy is 0.847 based on the F1 score.

- As a deep learning-based biometrics, the accuracy is lower than expected. The F1 score of 0.847 is insufficient to be used for 1:1 authentication.

- Image alignment process is not considered. There are geometric variation in the video captured by the camera. How do you solve this factor? As shown in the structure of the device in Figure 10, there is no guide that can control the geometrical variables of the wrist image.

- The vein region extracted from figure 9 seems to include the bracelet(?) region. Are these false positive issues not considered?

- The caption of Figure 4 is insufficient. Please, give sub-labels for the 3 images, and add sub-captions describing each one.

- The biometrics research results should present the accuracy in terms of FAR/FRR/EER. An ROC curve is also required. In a 1:1 matching scenario, it is necessary to present FRR when FAR is close to 0.

- Section 3.5 is general, not necessarily included in research papers. Consider exclusion.

Author Response

We acknowledge the valuable input from the reviewer. The comments are specific, targeted, and aids us to improve this article pushing it towards perfection so that it is useful for the wider scientific research community.

In this paper, the authors proposed a biometrics method using wrist vein patterns. The DB used for verification is FYO, the vein extraction accuracy is 0.723 based on the Dice correlation coefficient, and the recognition accuracy is 0.847 based on the F1 score.

  • As a deep learning-based biometrics, the accuracy is lower than expected. The F1 score of 0.847 is insufficient to be used for 1:1 authentication.

 Response:

We acknowledge this comment from the reviewer. The focus of this article is to present a fully functional end-to-end wrist vein biometric system methodology using deep learning. This work can be extended to other fields and will require some level of parameter optimisation. Most of the existing literature do not present  any working end-to-end system. This is the first work to the best of our knowledge where a reader can benefit from the  information in the article and successfully implement an entire system. Once the system is developed, it can always be optimized for higher F1 score. The focus here is on the methodology and deliver stepping-stone results to prove it is promising to adopt. Our future work will be targeted on improving the performance parameters by adjusting hyperparameters to make it more acceptable in real world applications. This has been highlighted in the conclusion and future works section of the article. 

- Image alignment process is not considered. There are geometric variation in the video captured by the camera. How do you solve this factor? As shown in the structure of the device in Figure 10, there is no guide that can control the geometrical variables of the wrist image

Response:

Thanks to raise this important issue to show the difference between the classical machine learning approaches and CNN. 

The primary aim was to develop a contactless wrist vein recognition system.  We specifically focused on the contactless element and have accounted for the issues that could be present due to misalignment of the wrist during the image acquisition process. Introducing guides to control the placement of the user's wrist has been proposed and used successfully in literature but would have defeated the goal of a contactless system. Geometric variations in the input image are accounted for by the Siamese and convolutional neural networks used in the matching stage. As shown by Goodfellow et al. in [1], the pooling layers in these networks have the ability to learn to account for these geometric variations. The dataset images used to train our model were augmented with translation, rotation and scaling to train the network to account for these factors. Alignment issues are one of the major issues to deal with in the classical machine learning modelling that depends on the feature extraction but not any more in the CNN modelling. This point is clarified in the new version of the paper.

[1]: I. Goodfellow, Y. Bengio, and A. Courville, Deep Learning. MIT Press, 2016

- The vein region extracted from figure 9 seems to include the bracelet(?) region. Are these false positive issues not considered?

Response:

False positive and false negative issues introduced by accessories such as jewellery were not specifically addressed during this article. The system was evaluated holistically, including cases where jewellery is present, as would be seen in the real world. The authors agree and acknowledge that an investigation surrounding the effects of jewellery would provide value. This variation is again considered with the help of data augmentation in the current experimentation stage. Definitely, this input from the reviewer is precious and can help the extension work in the future. A comment on this has been added to the current version of the manuscript.

- The caption of Figure 4 is insufficient. Please, give sub-labels for the 3 images, and add sub-captions describing each one.

Response:

We acknowledge this point raised by the reviewer. The captions have been reviewed and updated. Sub-figures are now independently annotated with appropriate naming.

  • The biometrics research results should present the accuracy in terms of FAR/FRR/EER. An ROC curve is also required. In a 1:1 matching scenario, it is necessary to present FRR when FAR is close to 0.

Response:

We acknowledge this point raised by the reviewer. A confusion matrix based analysis would be the standard protocol for biometrics research in the current research trands where the accuracy in terms of FAR/FRR/EER could be reported. This would further lead to generation of an ROC curve. Since the aim of this article was to present the novel biometric system as a whole, we have chosen to report the image matching performance with the help of accuracy, loss, Dice Coefficient and F1-score alone. Optimization of the developed models that can be tested successfully with this novel system is the next step of this work. At that stage, it would be ideal to perform a confusion matrix based analysis to report precise performance parameters of the biometric system. We only report the necessary parameters to ensure the highlight of the article is on the novelty of the complete end-to-end system. The metrics used are currently adopted in reporting system performances and used for comparison too. We again acknowledge this comment and will comment on this point in our updated version of this paper.

- Section 3.5 is general, not necessarily included in research papers. Consider exclusion.

Response:

We acknowledge this comment raised by the reviewer. Section 3.5 is intended to simply provide a high level introduction to the graphical user interface that will be made available to the readers on request. This, again, to the best of our knowledge, is unique as the implementation details and codes are being made available for the benefit of the wider research community. We take this comment from the reviewer positively and have made this section more precise and targeted. We prefer not to completely exclude the section as it may be beneficial to the readers who wish to implement/replicate and try using the end-to-end wrist vein vascular biometric system for further optimization.

Reviewer 3 Report

The article proposes Deep Learning Based Wrist Vascular Biometric Recognition. The article cannot be accepted in its current form. However, I propose to improve the article by addressing the following points:

1. More numerical/statistical results are to be added to the abstract.

2. The abstract needs to be improved by highlighting the novelty of the work.

3. The article seems to be an extension of the previous paper "Marattukalam, F.; Cole, D.; Gulati, P.; Abdulla, W.H. On Wrist Vein Recognition for Human Biometrics." In Proceedings of the 2022 Asia-Pacific Signal and Information Processing Association Annual Summit and Conference (APSIPA ASC). IEEE, 2022” and the amount of technical contribution is less.

4. Line 209 - “We developed a siamese neural network for matching palm vein patterns” To support the above statement, no literature survey is provided.

5. Line 253-256 “..was resizing to a resolution of 256 x 256. This was found to have been the best 255 tradeoffs between segmentation quality and time. Higher resolutions maintain more detail 256 but take longer to process”

To support the above statement, no literature survey is provided. The latest processors can easily implement it with a lesser time to process and this statement is not acceptable.

6. I did not find any information about the number of epochs used for network training. For any training network, the number of epochs is very significant.

7. There are no proper explanations provided about the number of layers used in the training and validation of results. Details like the number of layers, the number of neurons per layer, and the number of training iterations are to be added to the manuscript.

8. The author has used well-recognized and published methodologies with some modifications. There are no innovations to be found in this article.

9. Authors need to bring novelty and originality to their work and need to establish the clear superiority of their proposed methodology through comprehensive comparison results with very recent algorithms.

10. The contribution of the results section is very limited. In the proposed model, there is a lack of explanations in detail as to why better results are obtained.

11. What are the key parameters of the proposed method? How will changing them affect performance?

12. It is a relatively weak experimental study. More comparisons with other existing methods should be provided.

13. The authors said they used the HK PolyU palm vein dataset, but didn't talk about the subjects used in the data set in detail.

 14. The article seems to be a technical report on an image acquisition system and the authors need to compare their proposed model with existing models.

Author Response

The article proposes Deep Learning Based Wrist Vascular Biometric Recognition. The article cannot be accepted in its current form. However, I propose to improve the article by addressing the following points:

We acknowledge the valuable inputs from the reviewer to improve the article. Below is the detailed response to the points raised. The response reflects the changes that have been incorporated to improve article and also the summary to the key points raised by the reviewer.  

  1. More numerical/statistical results are to be added to the abstract.

Response:

We acknowledge this point raised by the reviewer. We have included more specifics into the abstract to ensure the extent of research is reflected in the abstract. 

  1. The abstract needs to be improved by highlighting the novelty of the work.

Response:

We acknowledge this point raised by the reviewer. To ensure the originality and novelty of work is better appreciated, we now have redrafted the abstract highlighting the achievements. 

  1. The article seems to be an extension of the previous paper "Marattukalam, F.; Cole, D.; Gulati, P.; Abdulla, W.H. On Wrist Vein Recognition for Human Biometrics." In Proceedings of the 2022 Asia-Pacific Signal and Information Processing Association Annual Summit and Conference (APSIPA ASC). IEEE, 2022” and the amount of technical contribution is less.

Response:

We would like to draw the reviewer’s attention to our previous paper "Marattukalam, F.; Cole, D.; Gulati, P.; Abdulla, W.H. On Wrist Vein Recognition for Human Biometrics." In Proceedings of the 2022 Asia-Pacific Signal and Information Processing Association Annual Summit and Conference (APSIPA ASC). IEEE, 2022”. This is a review paper and does not report any results whatsoever. The review carried out in this paper aided the development of this end-to-end deep learning based wrist vascular biometric system and that is the reason we have referenced this literature paper in this article. In no way this is an extension if the above said work. 

  1. Line 209 - “We developed a siamese neural network for matching palm vein patterns” To support the above statement, no literature survey is provided.

Response:

We acknowledge this point raised by the reviewer. The literature survey and study was performed in the reference paper provided in line 209 and the authors found it to be repetitive to provide these details in an article related to wrist vein recognition. We have deliberately avoided this repetition to ensure that we are precise and only reporting specifics.

  1. Line 253-256 “..was resizing to a resolution of 256 x 256. This was found to have been the best 255 tradeoffs between segmentation quality and time. Higher resolutions maintain more detail 256 but take longer to process”. To support the above statement, no literature survey is provided. The latest processors can easily implement it with a lesser time to process and this statement is not acceptable.

Response:

We acknowledge this point raised by the reviewer. The ‘longer to process’ statement is quite general and was intended to only provide a holistic view of the computational load. We have included literature in the modified version of the manuscript to ensure that this statement is well supported. We also have indicated that this was intended to give a overall picture of the system’s ability to perform on an ordinary processor. We understand that newer processors can digest more computational load easily, this would mean significant increase of system cost which would make this system less desirable. We accept the fact that the statement is misleading and have accordingly revised the same in the current version of the manuscript. 

  1. I did not find any information about the number of epochs used for network training. For any training network, the number of epochs is very significant.

Response:

We acknowledge this comment from the reviewer.. The number of epochs each network was trained for was provided in the current version of the article. For reference, these statistics can be found in the following sections: U-Net for image segmentation: Section 4.2, 107 epochs, Convolutional neural network for image matching: Section 4.3.1, 200 epochs, Siamese neural network for image matching: Section 4.3.2, 100 epochs. We have reworded these statements to ensure they are more readable and clear to the reader. 

  1. There are no proper explanations provided about the number of layers used in the training and validation of results. Details like the number of layers, the number of neurons per layer, and the number of training iterations are to be added to the manuscript.

Response:

We acknowledge this comment from the reviewer. The manuscript has now been updated to include the number of tunable layers and number of trainable parameters for each proposed network.  These details were not explicitly mentioned in a readable format and now have been included in the updated version of the manuscript..The number of training iterations (epochs) has already been included as mentioned in response for point 6.

  1. The author has used well-recognized and published methodologies with some modifications. There are no innovations to be found in this article.

Response:

We reiterate the fact that the main focus of this article is to report a fully functional end-to-end system using deep learning. This work can be extended to other fields. The models used in the development of the system are optimized versions of our own models developed for similar applications using a different modality. The innovative factor and the novelty is explicitly highlighted on page 2. We would like to draw the reviewers attention to the introduction and results section. We appreciate the valuable input from the reviewer and have accordingly improved the results and introduction section to ensure the core idea behind the article is clear to the reader. 

  1. Authors need to bring novelty and originality to their work and need to establish the clear superiority of their proposed methodology through comprehensive comparison results with very recent algorithms.

Response:

We acknowledge this comment from the reviewer. Most of the reviewed literature present comparative study and performance parameters alone. But, it’s impossible to verify the work mentioned as often the information provided is insufficient to develop the reported system. This article’s novelty lies in the fact that an entire functional wrist vein vascular biometric system with details of each stage is presented. To the best of our knowledge, there is no such work that reports all the systems, puts them together and presents it as a working system. This was the case for both commercial and academic literature. We have highlighted this in the literature review presented in this article. We agree with the reviewer on improving the claim to establish superiority over other system. But other systems are only sub systems, so to present a detailed comparison, we would need to do a performance reporting and comparison results for each sub system. This definitely would highlight the novelty of the work even more but is not in the scope of the current article. The current article is centered on reporting the entire system as a whole and is intended for the reader to be able to develop an end-to-end system.     

  1. The contribution of the results section is very limited. In the proposed model, there is a lack of explanations in detail as to why better results are obtained.

Response:

We would like to draw the reviewer's attention to page 2 (Introduction section). This section very specifically mentions the contribution of the paper. The results section is simply reporting the results and not highlighting the contributions of the paper. The proposed model, subsystems and stages are elaborately mentioned in sub-sections 3.1, 3.2, 3.3, 3,4 and 3.5. These subsections elaborate on the complete details of different stages that make the entire working system. We have tried out best to display this both visually and in terms of text. We have redrafted the results section to ensure more comprehensive coverage of the system and to ensure the reader is directed to the correct location of the article to appreciate the novelty and contributions of the work completed. 

  1. What are the key parameters of the proposed method? How will changing them affect performance?

Response:

If the reviewer is referring to the key parameters of the models proposed, these are mentioned in detail in subsection 3.4,  Image Matching and Decision Making Subsystem. When using deep learning based approaches, certain parameters types are common across models. Newer parameters maybe added depending on the optimization requirement. Changing the parameters may result in degradation or improvement of the model performance. At this stage, certain parameters were chosen and set when the system started to perform at an acceptable level. The acceptable level was when the system was able to identify an image, say for 1:1 matching with at least 80% accuracy. The focus of the future work would be to optimise this based on international IEC/ISO standards and then compare it with other deep learning ( if available) or non deep learning based methods reported in literature.   

  1. It is a relatively weak experimental study. More comparisons with other existing methods should be provided.

Response:

We acknowledge this input from the reviewer. This article does report an experimental study but we strongly believe it would be pivotal in the future research towards wrist vein vascular biometric systems. It is to be noted that this article reports all the stages involved in the development of a working deep learning based wrist vascular biometric system. So the focus of the article is to appreciate the novelty of the work and enable the reader to develop a similar system for further improvement or even commercial deployment. Based on the valuable input from the reviewer we now have added a few more comparisons into the study. But, it is to be noted that there are very few literatures that report all the aspects of such a system. This is a challenge we intend to approach as part of our future work.   

  1. The authors said they used the HK PolyU palm vein dataset, but didn't talk about the subjects used in the data set in detail.

Response:

We have used the PUT wrist vein database and not the HK PolyU database. The HK polyU multispectral palmprint and palm vein databases are as its mentioned ‘palm’ based databases. We did use the HK polyU database when we worked on palm vein recognition systems and this has been referenced in this article. This is the reason why extensive information about the HK polyU database is not mentioned in the article. Including this into the article would derail the readers focus on to palm and not serve the purpose of the article in its essence. We have referenced the work and also provided citation to HK polyU database to ensure comprehensive details of the database in available to the reader. 

  1. The article seems to be a technical report on an image acquisition system and the authors need to compare their proposed model with existing models.

Response:

The article is focused on introducing the novel work carried out for the development of  an end to end wrist deep learning based wrist vein recognition system.  Due to the need for reporting of the various segments that completes the system, it tends to present itself as a technical report. We acknowledge this input from the reviewer and have accordingly redrafted sections of the article so that it is more comprehensive. To the best of our knowledge, there are no other deep learning based wrist vascular biometric systems or literature that report and end to end working system. This is the reason the work is not extensively compared with other literature. The next stage of this work would be optimization and comparison with other research. This would be done by breaking the system down into stages and then comparing each of the stages with other available literature. We now have indicated this part into the paper. 

Round 2

Reviewer 2 Report

The title of this paper is "Deep Learning Based Wrist Vascular Biometric Recognition". However, the authors responded to my comment as follows.

"Since the aim of this article was to present the novel biometric system as a whole, we have chosen to report the image matching performance with the help of accuracy, loss, Dice Coefficient and F1-score alone."

Is the point of this paper "Recognition" or "System"?

If "System" is the key, the paper's contribution is weak, and if "Recognition" is the key, experiments and analysis are lacking.

It is advised to resubmit with exact targeting of the core.

Author Response

We thank the reviewer for the valuable comments and inputs to help us improve this article. We appreciate the time you have taken to provide us with a peer review. This peer review is helpful for us to address the core contributions of the paper and further highlight the novelty of the article. 

The following response addresses the points raised by the reviewer.

The title of this paper is "Deep Learning Based Wrist Vascular Biometric Recognition". However, the authors responded to my comment as follows.

"Since the aim of this article was to present the novel biometric system as a whole, we have chosen to report the image matching performance with the help of accuracy, loss, Dice Coefficient and F1-score alone."

Is the point of this paper "Recognition" or "System"?

If "System" is the key, the paper's contribution is weak, and if "Recognition" is the key, experiments and analysis are lacking.

Response

The key of this paper is the wrist vein recognition system developed as a whole and not recognition only. We acknowledge in such case the contributions may appear/seem weak. To ensure that is not the case, we have highlighted the contribution more explicitly in lines 81 to 92 of the paper. For simplicity, the same has been elaborated on below. 

The core contribution of the paper revolves around 1. Development of a fully functional low-cost contactless vein image acquisition system to capture high-quality vein pattern images and match them in a biometric system. 2. Detailed analysis and reporting of NIR wavelengths suitable for use in vein biometrics and specifically wrist vein biometrics. This is beneficial for the extension of this research to develop a more precise ‘Recognition’ system by performing future experiments and analysis easily. 3.  Image segmentation using the U-Net convolutional neural network proposed in one of our previous works. We report that this segmentation method with custom optimization for wrist images is successful and good enough to be integrated into a working system. 4. Image matching with siamese and convolutional neural networks as an extension of our previous work for palm vein recognition. Two neural networks are designed, developed, and reported. The parameters are explicitly mentioned in this article, making this article valuable to future researchers to expand this work.   5. Design and development of a GUI to interface with the system to make it reproducible for further testing. Without the interface, the system would not be user-friendly. We want interested researchers to benefit from this article by easily implementing it and then expanding the work to perfection. This opens room for the focus to shift to ‘recognition’ and hence the optimization of design decisions. We strongly believe that this contribution is significant, as to the best of our knowledge there has been no comparable literature of this sort. This system will also help in future works for the recognition/machine learning aspect of the system and can be pivotal in this field of research. 

Reviewer 3 Report

The authors address all the queries raised and I am satisfied with their responses.

Author Response

We thank the reviewer for the valuable input that helped us improve this article. 

Round 3

Reviewer 2 Report

It is acknowledged that clearly defining the contributions of the proposed method contributed to the great quality improvement of this paper.

But it raises one important new issue.

In the proposed method, U-Net is used to detect the blood vessel area.

Matching is performed by inputting the binary images of detected blood vessels in pairs to the Siamese network structure as shown in figure6.

In the convolutional neural network, the vein area is reflected as an important feature in the learning process of generating learned features. (If you trained the network properly...)

Detecting the blood vessel region through U-Net and learning by inputting the resulting image reduces the reason for adopting a CNN-based learning network to train the learned features.

I am curious about the performance when learning the input image of Figure6 through the original image rather than the binary image.

In my opinion, you can expect better performance.

Please change the words "Candidate" and "Known genuine" in Figure6 to "Input image" and "Enrolled image" respectively.

As a result of blood vessel detection, please present the genuine / imposter distribution for when a binary image is input and when an original image is input.

Author Response

We acknowledge the valuable inputs from the reviewer to improve the article. Outlining the contributions explicitly was something that we could improve due to these inputs. Thanks again.

It is acknowledged that clearly defining the contributions of the proposed method contributed to the great quality improvement of this paper.

Below are the detailed responses to the points raised. The responses reflect the changes that have been incorporated to further improve the article and also the summary to the key points raised by the reviewer. 

 But it raises one important new issue. In the proposed method, U-Net is used to detect the blood vessel area. Matching is performed by inputting the binary images of detected blood vessels in pairs to the Siamese network structure as shown in figure6.

In the convolutional neural network, the vein area is reflected as an important feature in the learning process of generating learned features. (If you trained the network properly...)

Detecting the blood vessel region through U-Net and learning by inputting the resulting image reduces the reason for adopting a CNN-based learning network to train the learned features.

I am curious about the performance when learning the input image of Figure6 through the original image rather than the binary image.

In my opinion, you can expect better performance.

As a result of blood vessel detection, please present the genuine / imposter distribution for when a binary image is input and when an original image is input.

 Response:

As rightly pointed by the reviewer, the U-Net helps with the generation of “mask” images from the original image i.e. image captured by the wrist vein scanner. If we were to use only a Convolutional Neural Network for the entire matching engine, U-Net should not have been of focus as the primary logic would be to optimize the CNN parameters to optimally pick the vein features from the original image itself. This would be the most logical approach in applying CNN and has already been attempted by us for Palm Vein images. As rightly suggested by the reviewer, the feature learning in this process was superior and we obtained better accuracy. In such case, the inputs shown in Figure 5 i.e. CNN Matching Network Architecture, should have been only the original image and not the mask image that was generated from the U-Net. However, showing that a complete CNN would also work was primarily to highlight that it was also an option to be considered by the readers in case where ample image data is available for training and balance it out with the computational resources needed for the same. However, the idea here was to bring the focus of the reader towards using Siamese Neural Networks where the sub networks would be taking the mask images (which are generated by the U-Net) and output 1 dimensional features for simple comparison. This is beneficial when the number of samples are very less and echoes with the concept of few shot learning. When applying Siamese Neural Networks, providing the original image instead of the mask was attempted, and the results were inferior in 1 to N matching scenario. This is the reason we opted to use the mask images obtained from the U-Net as it already showed superior results and could be viably supported with recent literature. Also, since Siamese neural networks work on similarity scores between different images, generating 1-D number between binary mask images were less computationally intensive and more accurate. This is also the reason that it can be run using TensorFlow lite on a Rasberry Pi 4. We can easily implement the system with the original images (actual or pre-processed version) and report the results in this paper. However, it would conflict with the core philosophy of working with few images (Few shot learning and Siamese networks), and also affect the performance analysis as such a system is heavy to be run on Rasberry Pi 4. We are confidently stating this because we have attempted this for palm vein images, and the entire matching engine could not be translated onto a Rasberry Pi 4. We hope we did justice in responding to this point from the reviewer, and clearly highlighted how this change would clash with the core focus of this paper.

Please change the words "Candidate" and "Known genuine" in Figure6 to "Input image" and "Enrolled image" respectively.

Response:

We acknowledge this input from the reviewer and also understand the reasons behind the same from a biometrics point of view. As such, the same has been incorporated in the updated version of his paper.

Round 4

Reviewer 2 Report

The authors adequately explained the reason for using the vessel mask images detected through U-Net as the input of the CNN.

However, unless this explanation is mentioned in the paper, the fact that binary images are used as input to CNNs will not convince readers.

It is recommended that you summarize your answers in appropriate sections of the manuscript so that readers can understand them.

Author Response

The authors adequately explained the reason for using the vessel mask images detected through U-Net as the input of the CNN.

However, unless this explanation is mentioned in the paper, the fact that binary images are used as input to CNNs will not convince readers.

It is recommended that you summarize your answers in appropriate sections of the manuscript so that readers can understand them.

Response:

We acknowledge this valuable input from the reviewer and are also glad that we were able to adequately explain the reasons behind using vessel mask images detected using our U-Net architecture. We have cross-referenced Subsection 3.4.2 with Section 3.3 and explicitly mentioned the reasons behind using U-Net for mask image generation. This is to help convince the readers to appreciate the difference between the use of original images and mask images as input to the Siamese Neural Network. The updated version of the manuscript encompasses this change.